# Complementary and Integrative Management of Pediatric Lower Urinary Tract Dysfunction Implemented within an Interprofessional Clinic

**DOI:** 10.3390/children6080088

**Published:** 2019-07-30

**Authors:** Kathryn E. Morgan, Susan V. Leroy, Sean T. Corbett, Jaclyn A. Shepard

**Affiliations:** 1Department of Pediatric Urology, University of Virginia, P.O. Box 800422, Charlottesville, VA 22908, USA; 2Department of Psychiatry and Neurobehavioral Sciences, University of Virginia School of Medicine, P.O. Box 800223, Charlottesville, VA 22908, USA

**Keywords:** lower urinary tract dysfunction, dysfunctional voiding, bladder and bowel dysfunction, constipation, interprofessional, movement therapy, body awareness, biofeedback, pelvic floor dysfunction

## Abstract

Lower urinary tract dysfunction in children is a common multifactorial functional problem that often correlates with bowel dysfunction and behavioral disorders. Ideal management combines integrative therapies that optimize bladder and bowel habits, address behavioral issues, foster mind–body connection, and improve pelvic floor muscle dysfunction. Movement therapies that teach diaphragmatic breathing and relaxation, mind–body awareness, and healthy pelvic floor muscle function are vital for long-term symptom improvement in children. This paper outlines recommendations for integrative management of these patients and discusses a recently developed interprofessional clinic that aims to better meet these patients’ complex needs and to provide patients with an integrated holistic plan of care. Additional work is needed to scientifically assess these treatment models and educate providers across the various disciplines that evaluate and treat these patients.

## 1. Background

The National Institutes of Health National Center for Complementary and Integrative Health describes integrative medicine as the practice of bringing together conventional and complementary, or non-mainstream, interventions to provide well-coordinated and holistic care of a whole person [1]. This approach towards health care is beneficial to a wide variety of patients, including children with lower urinary tract dysfunction (LUTD). Integrating the use of movement, mind–body techniques, and dietary modifications is essential to the long-term improvement in this population.

The International Children’s Continence Society (ICCS) defines LUTD as the presence of irritative voiding symptoms in a patient age five or older over a given period of time specific to the symptom. The most common symptoms are increased urinary frequency, urinary urgency, daytime urinary incontinence, and dysuria, but also include any symptom listed in Table 1. LUTD symptoms can occur during any phase of micturition including bladder filling, storage, or active voiding. To be diagnosed with LUTD, a patient may only have one symptom, but often patients have multiple, overlapping lower urinary tract symptoms [2].

There is wide variability in the clinical presentation of LUTD due to the myriad of associated symptoms. LUTD can contribute to recurrent urinary tract infections, vesicoureteral reflux (VUR), and resultant renal scarring. It is considered a negative prognostic indicator of spontaneous resolution of VUR [3]. Additionally, children with LUTD often have parallel bowel dysfunction. In fact, studies estimate that up to 50% of children with LUTD also have functional constipation by the ROME III criteria for diagnosing defecation disorders in children [4]. The ICCS defines the presence of these two functional issues together as bladder and bowel dysfunction (BBD) [2]. 

Psychological aspects of BBD are generally understudied, though there is growing evidence to suggest that children with these conditions and their parents experience a lower quality of life and increased distress compared to healthy controls and those with other chronic GI medical conditions [5,6,7,8,9,10]. Peer teasing and rejection secondary to incontinence can result in negative self-esteem, learned helplessness, or lower social functioning [11,12]. More recent findings also suggest high rates of concomitant developmental delays and mental health conditions, such as attention deficit/hyperactivity disorder, oppositional defiant disorder, anxiety, and depression [10,13]. In fact, children with functional incontinence have rates of behavioral comorbidities that are three to six times higher than those without incontinence [14]. Although subclinical psychosocial symptoms tend to remit with improvements in bladder and bowel functioning, clinically significant behavioral and emotional disorders are more pervasive and are more likely to impact treatment adherence and overall outcomes [15]. Therefore, integrating early screening, education, and treatment of underlying psychological conditions into medical treatment protocols is recommended, particularly in consideration of the biobehavioral framework of LUTD [16]. 

The etiology of LUTD is multifactorial, with a possible maturational delay of the lower urinary tract. However, in an otherwise healthy child, the condition is most often functional in nature without any physiological or anatomical origins [17]. To treat the underlying cause of this condition, it is indicated to begin with non-medical interventions that address the functional basis of the problem. It is well established that integrative therapies, including movement based interventions, are the optimal first line treatment of LUTD. Patients may also need supplemental medical therapy to address persistent symptoms or treat infections, but it has also been shown that medical therapy will rarely change the underlying cause of LUTD without the use of behavioral interventions and alternative therapies. Furthermore, some medical interventions may complicate the dysfunction and cause even more urinary symptoms. For example, giving anticholinergics to control urinary frequency or urgency can lead to incomplete bladder emptying and exacerbate bowel dysfunction by increasing constipation. 

Up to 40% of all patients who present for pediatric urology visits meet the criteria for LUTD [18]. Given the multifaceted nature of evaluation and treatment of these patients, an environment that facilitates a holistic and comprehensive approach to patient care is optimal. At our institution, we have recently designed and implemented an interprofessional clinic that allows for a more comprehensive evaluation of a patient’s LUTD that includes ruling out any underlying medical pathology, assessing the functional aspects of LUTD and BBD, and addressing associated behavioral, mind–body, and movement based needs.

## 2. Complementary and Integrative Interventions for Children with LUTD

### 2.1. Urology Interventions

In children with urinary incontinence, recurrent urinary tract infections, and irritative voiding symptoms, modifications to voiding and hydration habits are imperative to improvement. Children often postpone voiding or do not attune to bladder sensations indicating the correct time to void [19]. Chronic holding of urine can lead to bladder overdistension, which can cause changes to the detrusor muscle [20] and alter bladder sensations about the appropriate time to void. These behaviors and accompanying bladder changes can contribute to a myriad of problems, including overflow incontinence, urge incontinence, incomplete bladder emptying, and increased risk for urinary tract infections [21].

It is often recommended that patients begin a timed voiding schedule, where they attempt to void every two to three hours regardless of the sensation to void. Watches that can be set to vibrate at a specific time interval can be useful for reinforcing this behavior [22] along with reward systems for performing the behavior. The practice of double voiding can help patients with incomplete bladder emptying. This process requires patients to void, step away from the toilet for several minutes, and then attempt to void again. Adequate fluid intake with beverages that are not bladder irritants (i.e., caffeine, citrus, chocolate, and carbonated beverages) is also crucial for improvement of lower urinary tract symptoms [23], including bladder urgency, frequent urination, and dysuria. It is helpful to educate families on the practical ways to achieve this with children, including offering flavored water, creating water goals to achieve throughout the day, and apps to track water intake. Parents must also understand that altering their own habits can have a significant positive or negative impact on their child’s habits.

Addressing hygiene is another important component of improvement in LUTD. Patients are instructed on the proper wiping method to prevent bringing stool up to the urethra. Other practical strategies for patients, especially those with perineal irritation, include wearing cotton underwear, changing out of wet underwear, and sitz baths with baking soda. Additionally, some girls reflux urine into their vagina that will dribble out soon after standing from voiding. This can typically be remedied by sitting on the toilet with knees spread wide or sitting backwards on the toilet and opening the labia before and after voiding as well as making sure underwear is pulled down below the knees [24].

The use of a toileting diary can be helpful for patients and families to identify the child’s day-to-day bowel and bladder habits and to identify specific areas for modification and improvement. We must note that the success of these behavior modifications often depends on the commitment of all of the child’s caregivers, daycare providers, and school personnel to reinforce these habits on a daily basis.

### 2.2. The Importance of the Pelvic Floor

Urine storage and voiding is a complex process involving intricate communication between the brain, spinal cord, bladder, and pelvic floor musculature. Some children with lower urinary tract symptoms have normal voiding mechanics whereas others have true ‘dysfunctional voiding,’ meaning that the urinary sphincter and pelvic floor musculature is contracted instead of relaxed during voiding [2]. Dysfunctional voiding can cause changes to the detrusor (bladder) muscle, which can potentiate incomplete bladder emptying, incontinence, urinary tract infections, and vesicoureteral reflux [21]. In our experience, we find that pelvic floor dysfunction in children often correlates with a variety of factors including painful voiding and stooling, anxiety and life stressors, and history of sexual abuse or other trauma. The presence of pelvic floor muscle dysfunction in these children has not been well studied specifically but there is evidence that prevalence of LUTD is higher in children who have experienced sexual abuse [25]. 

We can assess pelvic floor function in children with uroflow tests with or without pelvic floor muscle EMG measurement, measurement of post void residuals, and with more invasive urodynamic testing. Children with normal voiding mechanics will have a bell shaped uroflow curve; children with dysfunctional voiding will have a staccato/intermittent or prolonged uroflow curve [21]. Collecting a thorough history on voiding/defecation complaints, sensations, and habits along with current emotional and behavioral concerns is also important in garnering insight into possible pelvic floor muscle dysfunction.

Treating pelvic floor dysfunction is vital to the improvement of LUTD in children. Toilet positioning is an essential component of relaxation of the pelvic floor during voiding and stooling. Children should be taught to sit on the entire toilet seat with legs and feet supported and relaxed [26]. We often encourage families to use stools or other similar household objects at the base of the toilet for feet support and recommend that children avoid squatting at public toilets. Occupational therapists (OT) and physical therapists (PT) can help patients attune to body awareness and relaxation strategies. Therapists with an understanding of pelvic floor dysfunction are vital to assessing pelvic floor tone, strength, and flexibility and teaching children strategies for stretching, strengthening or relaxing the muscles depending on the child’s individual needs. Biofeedback therapy is a complementary training method for children with bladder and/or bowel dysfunction to learn how to tighten and relax their pelvic floor muscles while on the toilet. It has been shown in multiple studies to improve and/or correct pelvic floor dysfunction in children with LUTD [27].

### 2.3. Gastroenterology Interventions

It is well established that children with LUTD often have concurrent bowel dysfunction, including functional constipation diagnosed by the Rome III criteria. There are several theories about why this occurs, including pressure on the bladder from a large rectal stool burden causing bladder irritation and urethral obstruction, sharing of neural input from the bladder, bowel and corresponding sphincters, and stool withholding leading to concurrent difficulty relaxing the urinary sphincter [4]. Successful management of functional constipation involves a bowel cleanout as needed followed by a maintenance laxative regimen and a toilet sitting routine [28]. It is often suggested that dietary changes such as increased water and fiber intake are important for optimal gut health; however, there is little empirical evidence that these changes improve functional constipation [29]. There is some early evidence that biofeedback is a useful complementary treatment for functional constipation and associated encopresis compared to medication-only interventions [30]. More recent studies, however, have demonstrated mixed results as to whether there is additive benefit to medical or behavioral interventions, so additional studies are warranted to further clarify the clinical utility of this training [12]. Reflexology massage, where gentle pressure is applied to specific reflex zones that are thought to correspond to various organs and systems throughout the body, has been proposed as another complementary therapeutic approach for pediatric functional constipation and encopresis. There has been only one pediatric observational study to date on this approach and, although the results were positive (i.e., children experienced a significant reduction in fecal accidents and a significant increase in the frequency of BMs based on parent report), further empirical investigation is warranted [31].

### 2.4. Behavioral Interventions

Behavioral interventions targeting pelvic floor muscle dysfunction, such as biofeedback, are considered the first-line treatment for patients with LUTD [32]. These conservative interventions aim to increase the patient’s awareness of the function of the pelvic floor muscles and to promote increased muscle coordination, which ultimately help to correct dysfunctional voiding patterns and increase the patient’s ability to suppress urgency. Behavioral therapy is also an effective office-based intervention for urinary urgency and frequency, as well as bladder training. Behavioral therapy is typically provided by a psychologist or provider with equivalent training in behavioral modification. Common behavioral interventions include teaching relaxation strategies to the child (e.g., diaphragmatic breathing or “belly breathing,” progressive muscle relaxation) and providing behavioral parent training, such as creating progressive voiding schedules, monitoring voiding patterns via bladder diaries, and implementing distraction strategies with their children as needed. Younger children and/or youth with comorbid developmental or psychological conditions, such as ADHD, may benefit from having an adjunctive reward system in place to increase motivation and adherence, and ultimately self-awareness, through positive reinforcement. Cognitive behavioral therapy may be warranted to treat prominent comorbid psychological conditions, such as anxiety/phobias or depression [12]. 

For bowel dysfunction, enhanced toilet training (ETT) is an evidence-based, combined medical-behavioral intervention for constipation and encopresis that pairs laxative therapy with behavioral modification techniques, education, and training about appropriate defecation dynamics, including modeling of breathing techniques, abdominal muscle straining, and relaxation of the pelvic floor muscles [33]. As with behavioral interventions for LUTD, ETT incorporates daily diaries to monitor toileting patterns and contingency-based rewards to reinforce adaptive defecation dynamics, motivation, and adherence [16].

## 3. An Interprofessional Integrative Clinic Model

At our institution, we have recently developed a weekly interprofessional clinic with a pediatric urology nurse practitioner (NP), pediatric gastrointestinal nurse practitioner, and doctoral-level licensed clinical psychologist to evaluate and manage patients with concomitant LUTD, BBD, and associated behavioral needs. Two occupational therapists are peripherally involved to provide more intensive body awareness and relaxation techniques for select patients (i.e., yoga groups for adolescents). Patient referrals to pediatric urology for LUTD are initially reviewed by a triage RN, and the patients with indications of concurrent constipation and/or behavioral concerns are scheduled in the interprofessional clinic. On the day of clinic, initial visit patients have a renal ultrasound, uroflow, post-void residual, urinalysis, and abdominal x-ray prior to seeing all three providers for diagnostic interviews and physical examinations by the nurse practitioners. Follow-up visits are based on individual patient needs and may be scheduled in our interprofessional clinic or in the respective subspecialty clinic. Youth requiring ongoing mental health follow-up are either followed by our clinical psychologist for therapy, referred to local mental health providers closer to their homes, or referred for more comprehensive psychological evaluation when indicated. Due to a limited number of spots in our interprofessional clinic, the pediatric urology nurse practitioners continue to see many patients with LUTD in the regular medical/surgical clinic with similar initial visit testing and altered time slots to allow for longer, more comprehensive visits. 

Many of our patients could benefit from integration of movement therapy into their management and we commonly refer patients to outside physical and occupational therapists to provide these services. Unfortunately, access to this additional care is often limited in our region due to a lack of providers that can address pediatric PT/OT needs, transportation difficulties, and insurance coverage. We are currently working to facilitate education for regional providers and add additional therapy services at our institution to address these barriers.

## 4. Patient Examples

The following case examples demonstrate how the integration of movement based therapy and behavioral interventions into the plan of care for patients with LUTD can be helpful for symptom improvement. 

### 4.1. Example 1—Pelvic Floor Physical Therapy 

8-year-old female with recurrent urinary tract infections, urinary incontinence, and constipation. Figure 1 shows the uroflow from her initial visit in 2015, which indicates dysfunctional voiding and pelvic floor dysfunction. Her repeat uroflows continued to show this dysfunctional pattern despite initial urological interventions, as evidenced by Figure 2. Between the visits where Figure 2 and Figure 3 were obtained, the patient had one visit with an OT at our institution and eight visits with a local pelvic floor PT. Follow-up uroflow demonstrated improvement with an organized and bell-shaped curve. Although it is still somewhat unstable, it is less prolonged and staccato in shape. This patient told the practitioner she was “doing the peeing exercises they taught me”. 

### 4.2. Example 2—Pelvic Floor Physical Therapy

9-year-old female with urinary urgency, daytime urinary incontinence, and encopresis with complex medical and developmental history, including autism spectrum disorder, ADHD, sensory processing disorder, hypermobility, hypotonia, and fecal incontinence. Her original uroflow was staccato and interrupted, indicative of dysfunctional voiding. She had one visit with the OT at our institution and eight visits with a local OT for pelvic floor and body awareness. She was also instructed in enhanced toilet training techniques for bowel dysfunction. At follow-up 6 months later, urinary and fecal incontinent episodes had resolved. The guardian said the patient “did not recognize the feeling of the need to pee nor that she had muscles to control it, but once she got it, things got much better.”

### 4.3. Example 3—Behavioral Interventions

4-year-old female with extreme urinary frequency and a psychiatric history significant for anxiety. Her uroflow showed a staccato and interrupted pattern consistent with dysfunctional voiding. Patient reported experiencing symptoms of anxiety around urinary accidents, which led her to sit on the toilet excessively throughout the day, even at times when her bladder was just emptied. Psychological intervention included extensive psychoeducation about the physiological and behavioral aspects of anxiety that maintain or exacerbate urinary symptoms, modeling of/in vivo implementation of diaphragmatic breathing and distraction techniques with the child, as well as behavioral parent training to promote prolonged urine holding through the reinforcement of successive approximations. At patient’s first urology follow-up visit 4 months later, daytime frequency had resolved following these behavioral interventions. The patient continued to have a staccato, interrupted uroflow, but was beginning PT to address increased pelvic floor tone.

### 4.4. Example 4—Behavioral Interventions

7-year-old female with urinary urgency, frequency, daytime urinary incontinence, and nocturnal enuresis. Psychological history was significant for chronic worry related to urinary symptoms and in general. On her initial visit, she had a normal uroflow and moderate post void residual. The urology NP discussed double voiding, fluid intake, toileting posture, and management strategies for nocturnal enuresis. The clinical psychologist provided psychoeducation about the physiological aspects of anxiety and modeled and practiced diaphragmatic breathing with the patient to help reduce these sensations. Principles of cognitive behavioral therapy were also outlined, specifically the way in which thoughts, emotions, and behaviors influence each other. The psychologist demonstrated to the patient using a pertinent example how anxious thoughts, coupled with the physiological symptoms of anxiety, maintain and exacerbate urinary urgency and frequency. Behavioral parent training was also provided (e.g., perceived control through forced choices) to help reduce the patient’s general anticipatory anxiety. The patient followed up 5 months later and reported implementing recommendations made by the urology NP and the clinical psychologist with significant benefits, specifically, complete resolution of daytime incontinence, urgency, and frequency and significant reduction in nocturnal enuresis episodes from several nights a week to three over the past 5 months. The family attributed much of the patient’s progress to the cognitive behavioral interventions targeting the patient’s general anxiety.

## 5. Conclusions

Successful management of LUTD in children requires thoughtful holistic care that directly addresses urologic concerns but also addresses any concurrent bowel dysfunction and behavioral problems. Movement based therapies and behavioral interventions play a very important role in the treatment of LUTD. These integrative techniques that address behavioral issues, optimize elimination habits, improve control of the pelvic floor and overall sensory integration should be first line clinical interventions for LUTD. This paper outlines comprehensive recommendations for management of pediatric LUTD with a focus on complementary and integrative therapies and highlights how an interprofessional clinic model can better address these patients’ complex needs. 

### Future Directions

Based on the multifactorial nature of LUTD, integrative treatment approaches appear necessary, but there is limited empirical data on efficacy of interprofessional complementary and alternative interventions for pediatric LUTD. More feasibility studies are warranted to systematically assess multicomponent treatment models, including those which incorporate complementary and movement-based interventions. With more evidence, these models can be more widely implemented. In addition to scientific investigation, dissemination of complementary and integrative approaches to treatment of children with LUTD will require increased efforts in education across disciplines, particularly for primary care providers who are typically the first providers to evaluate these complaints, as well as increased interprofessional collaboration among providers caring for these patients. 

## Figures and Tables

**Figure 1 children-06-00088-f001:**
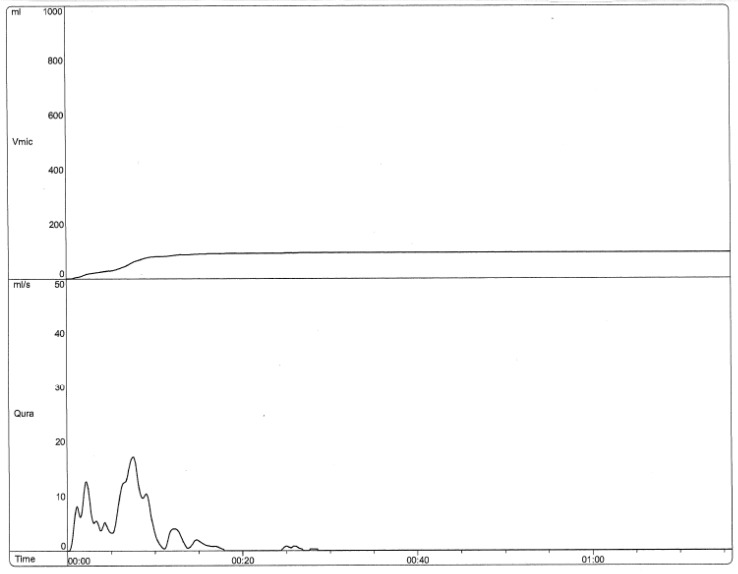
Uroflow (2015) Initial visit to clinic (Staccato and prolonged).

**Figure 2 children-06-00088-f002:**
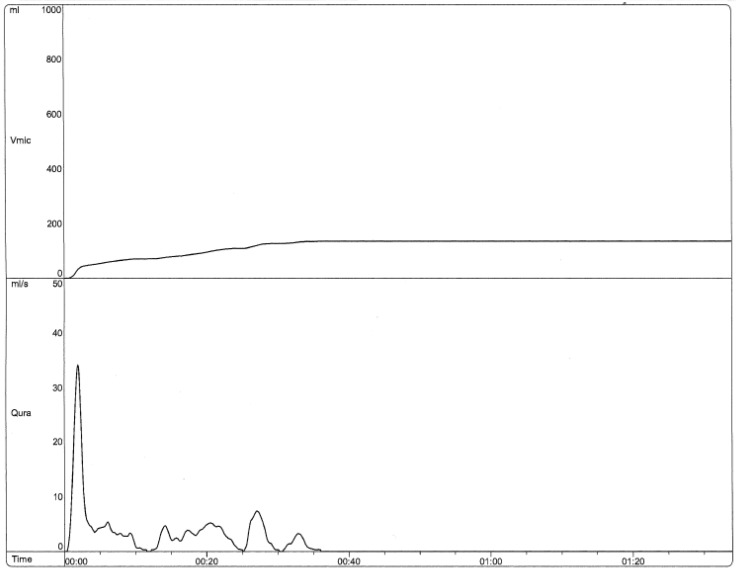
Uroflow (6/2018) Post 3 years of “traditional” bladder retraining without addition of movement therapies (Staccato and prolonged).

**Figure 3 children-06-00088-f003:**
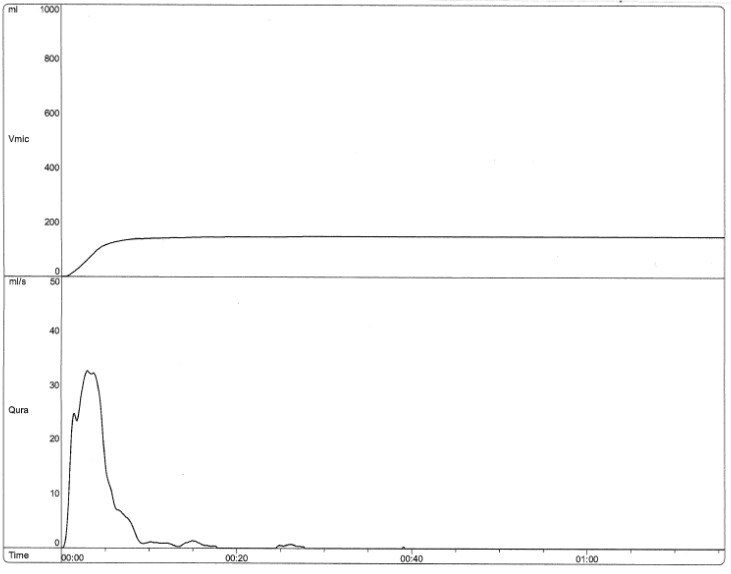
Uroflow (12/2018) Post 8 visits to Pelvic Floor Physical Therapist (bell shaped with some terminal dribbling).

**Table 1 children-06-00088-t001:** ICCS Lower Urinary Tract Symptoms.

ICCS Lower Urinary Tract Symptoms
Increased or decreased frequency
Incontinence
Urinary urgency
Nocturia
Hesitancy
Straining
Dysuria
Weak stream
Intermittent stream
Holding maneuvers
Feelings of incomplete emptying
Urinary retention
Post micturition dribbling
Spraying or splitting of the urinary stream
Pain of the LUT—genital, urethral or bladder

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
