# Peer review of "Complementary and Integrative Management of Pediatric Lower Urinary Tract Dysfunction Implemented within an Interprofessional Clinic"

_children, 2019, doi:10.3390/children6080088_

Round 1
Reviewer 1 Report
The authors are addressing an incredibly challenging and important topic. An integrating approach to pelvic floor dysfunction is thought to be the best way to help patients but one of the most difficult settings to set up an maintain in practice.
Comments:
"Given the prevalence of LUTD across the general pediatric population" - what is the prevalence? You discuss bowel dysfunction but do not actually state LUTD prevalence but you do mention that 40% of your patients who present for pediatric urology have LUTD.
Do you have any data about lag time, compliance, patient satisfaction for the "previous state"?
what ages of patients are you working with?
are these patients assessed by an doctor at any point prior to entering the clinic? If no, are they ever seen by a doctor if they continue in the clinic?
Do the providers review the cases as a whole to determine best care plan or does each individual provider give their own recommendations?
Typically how many patients can you see in one clinic? Ie. what is the realistic volume that can be seen? What is your wait time? You mention it is long - how long?
How many patients have you seen? Do you have outcome measures that you are currently tracking and assessing?
The design of your clinic is impressive. The patient examples are interesting, but they do not demonstrate necessarily how the integrative model of your clinic was used to achieve the patient goals. The same outcomes could have been achieved without your clinic model, albeit not likely as efficiently (but there is no information or data to demonstrate this). I am not sure the examples are beneficial to support your primary endpoint which is to improve patient access to care.
The conclusions that you draw are also not supported by any data. You describe this in your limitations.
Ultimately this could be a great study, if you had data to show that your clinic model is improving patient access to care. At this point you are only describing a model that you have developed and giving examples of how patients are being treated. Though this information is useful to anyone looking to develop a similar clinic, the lack of detail and outcomes data limits the utility of this paper.
Author Response
Thank you so much for reviewing our paper. Our main aims in this paper are two-fold: 1) to provide an overview of integrative management of children with LUTD, particularly for the urology naïve reader and 2) discuss specifically how movement based and behavioral therapies benefit these patients. This topic was requested specifically by one of the guest editors for the special edition of this journal focusing on how movement based therapies can be integrated into care of children with complex medical and behavioral needs. In reviewing your thoughtful comments and comments from the other review, we recognized that the in depth discussion of the development of our interprofessional clinic is a deviation from the aforementioned aims of this paper. Therefore, to reduce confusion for the reader and to preserve our original aims, we made significant revisions to exclude the details related to the development and implementation of our clinic and, instead, use our interprofessional clinic as a model of how to address the complex needs of children with LUTD.We believe this revision should address most of your concerns noted below, and we have also addressed additional feedback below.
The authors are addressing an incredibly challenging and important topic. An integrating approach to pelvic floor dysfunction is thought to be the best way to help patients but one of the most difficult settings to set up an maintain in practice.
Comments:
"Given the prevalence of LUTD across the general pediatric population" - what is the prevalence? You discuss bowel dysfunction but do not actually state LUTD prevalence but you do mention that 40% of your patients who present for pediatric urology have LUTD. This part of the sentence was removed. There is significant variability in the literature about what the prevalence of LUTD in the general pediatric population may be.
Do you have any data about lag time, compliance, patient satisfaction for the "previous state"? See above
what ages of patients are you working with? See above
are these patients assessed by an doctor at any point prior to entering the clinic? If no, are they ever seen by a doctor if they continue in the clinic? See above
Do the providers review the cases as a whole to determine best care plan or does each individual provider give their own recommendations? See above
Typically how many patients can you see in one clinic? Ie. what is the realistic volume that can be seen? What is your wait time? You mention it is long - how long? See above
How many patients have you seen? Do you have outcome measures that you are currently tracking and assessing? See above
The design of your clinic is impressive. The patient examples are interesting, but they do not demonstrate necessarily how the integrative model of your clinic was used to achieve the patient goals. The same outcomes could have been achieved without your clinic model, albeit not likely as efficiently (but there is no information or data to demonstrate this). I am not sure the examples are beneficial to support your primary endpoint which is to improve patient access to care. Introductory wording about examples was changed to show that examples reflect how integration of movement based and behavioral therapies are beneficial to patients with LUTD and aren’t necessarily specific to the clinic we have formed.
The conclusions that you draw are also not supported by any data. You describe this in your limitations.
Ultimately this could be a great study, if you had data to show that your clinic model is improving patient access to care. At this point you are only describing a model that you have developed and giving examples of how patients are being treated. Though this information is useful to anyone looking to develop a similar clinic, the lack of detail and outcomes data limits the utility of this paper. We appreciate this feedback and agree. As mentioned above, the details related to the feasibility of our interprofessional approach to this patient population is a deviation from the main aims of this paper and have been removed.
Reviewer 2 Report
Review for Manuscript children-529500-peer-review-v1
General Comments:
Initially, the manuscript was extremely well written and comprehensive.
Here are two other thoughts:
1) I think the manuscript should not be a “Review” paper, and should rather have the title changed from Complementary and Integrative Management of Pediatric Lower Urinary Tract Dysfunction Employed Within the Development of an Interprofessional Clinic” to "Complementary and Integrative Management of Pediatric Lower Urinary Tract Dysfunction Employed Within the Development of an Interprofessional Clinic: Background and Preliminary Evidence”. This is based on the fact they have an extensive background on the subject and then discuss their approach and summarize 4 cases.
2) Line 306-307 - For "Presently there are no formal outcome metrics available from this clinic, but our initial data is promising”, they should discuss more in the later “Future Directions” section how they plan (or propose how to plan) to design these metrics and future prospective trials to demonstrate the efficacy of their approach.
More Specific Comments:
Title – None
Abstract – None
Background
1) Line 41 – Change “is” to “are”
Body
1) Line 161-162 – Remove one use of “in children”
2) Line 249 – Remove the extra space after “they”
3) Line 400 – Remove underline before “Currently”
Tables – None
Figures and Legends – None
Author Response
Thank you so much for reviewing our paper. Our main aims in this paper are two-fold: 1) to provide an overview of integrative management of children with LUTD, particularly for the urology naïve reader and 2) discuss specifically how movement based and behavioral therapies benefit these patients. This topic was requested specifically by one of the guest editors for the special edition of this journal focusing on how movement based therapies can be integrated into care of children with complex medical and behavioral needs. In reviewing your thoughtful comments and comments from the other review, we recognized that the in depth discussion of the development of our interprofessional clinic is a deviation from the aforementioned aims of this paper. Therefore, to reduce confusion for the reader and to preserve our original aims, we made significant revisions to exclude the details related to the development and implementation of our clinic and, instead, use our interprofessional clinic as a model of how to address the complex needs of children with LUTD. We believe this revision should address most of your concerns noted below, and we have also addressed additional feedback below.
Review for Manuscript children-529500-peer-review-v1
General Comments:
Initially, the manuscript was extremely well written and comprehensive.
Here are two other thoughts:
1) I think the manuscript should not be a “Review” paper, and should rather have the title changed from Complementary and Integrative Management of Pediatric Lower Urinary Tract Dysfunction Employed Within the Development of an Interprofessional Clinic” to "Complementary and Integrative Management of Pediatric Lower Urinary Tract Dysfunction Employed Within the Development of an Interprofessional Clinic: Background and Preliminary Evidence”. This is based on the fact they have an extensive background on the subject and then discuss their approach and summarize 4 cases. This is a good point, and we have changed the title to: Complementary and Integrative Management of Pediatric Lower Urinary Tract Dysfunction Implemented Within an Interprofessional Clinic
2) Line 306-307 - For "Presently there are no formal outcome metrics available from this clinic, but our initial data is promising”, they should discuss more in the later “Future Directions” section how they plan (or propose how to plan) to design these metrics and future prospective trials to demonstrate the efficacy of their approach. See above
More Specific Comments:
Title – None
Abstract – None
Background
1) Line 41 – Change “is” to “are” The sentence reads as follows: “This approach towards health care is beneficial to a wide variety of patients…” “This approach” is singular and should be used in conjunction with “is” which is also singular.
Body
1) Line 161-162 – Remove one use of “in children” Done.
2) Line 249 – Remove the extra space after “they” Done.
3) Line 400 – Remove underline before “Currently” Sentence was removed.
Tables – None
Figures and Legends – None